# CoPhyBench: Benchmarking Physical Reasoning from Conditional Video Observation

## Abstract

We present CoPhyBench, a COnditional reasoning PHYsics-based BENCHmark. CoPhyBench evaluates the ability of Video-LLMs to reason about physical events based on conditional observations from real-world videos. It probes physics understanding from three perspectives: 1) Prediction: predicting future events from observable cues, assessing a grasp of causality in real-world scenarios. 2) Physical Calculation: estimating times and positions by translating visual conditions into variables of dynamics equations. 3) Counterfactual Reasoning: inferring futures based on hypothetical changes, to distinguish between generalizable physical understanding instead of superficial correlations. We construct a high-quality dataset consisting of 1,300 carefully verified question-answer pairs grounded in 232 diverse, real-world physics videos to support these tasks, spanning various phenomena in kinematics and dynamics. Extensive benchmarking on leading Video-LLMs reveals that while models perform reasonably on causal prediction, they struggle with precise physical calculations and counterfactual reasoning. These findings highlight the limitations of current models in transitioning from semantic alignment to deeper, physics-grounded reasoning, calling for new training paradigms to incorporate physics reasoning. Our dataset and resources will be released.

## 1 Introduction

Recent advances in large video–language models (Video-LLMs) have dramatically expanded real-world applications by leveraging data-driven methods that align visual content with language. However, they still struggle to ground visual observations with the underlying physical principles of our real world. Recently, physics-related benchmarks (Zhang et al., 2025b; Motamed et al., 2025; Zhang et al., 2025a) have emerged in response to the community's growing need to evaluate models' capability of reasoning based on physics knowledge. For example, (Motamed et al., 2025; Zhang et al., 2025a) explore how well video-generation models as world simulators comply with physical laws. Yet, without the clear basics of grasping real-world physical laws, video generation is disentangled from understanding. PhysBench (Chow et al., 2025) is the first comprehensive dataset to evaluate Video-LLMs' understanding of the physical world. However, as our benchmarking shows, it suffers from language biases inherent in common-sense knowledge of the LLMs. Thus, it is still unclear how well Video-LLMs use visual understanding to reason with physical knowledge.

To bridge this gap, we introduce CoPhyBench, a COnditional reasoning Physics-based BENCHmark. CoPhyBench probes Video-LLMs' understanding of physics from three perspectives: 1) Causal Prediction: Models are asked to predict future events based on conditional observations. Here, the model must infer from visual cues factors such as motion direction to anticipate physically plausible outcomes. 2) Physical Calculations: with stated assumptions, parse observed visual conditions into variables of physical equations, to make numerical predictions of timings and positions. 3) Counterfactual Reasoning: interpret changes based on hypothetical counterfactuals, to distinguish based on generalizable physical principles rather than superficial correlations. In the evaluation protocol, each video is segmented into two parts: a conditioning clip that presents the initial setup, and an event clip that contains the outcome, separated by a switch frame. The model is only given access to the condition segment. It is then prompted to predict what will happen next, estimate quantitative values such as the timing or location of an event, and reason about how changes in initial conditions would lead to different outcomes. This design allows us to assess whether mod-

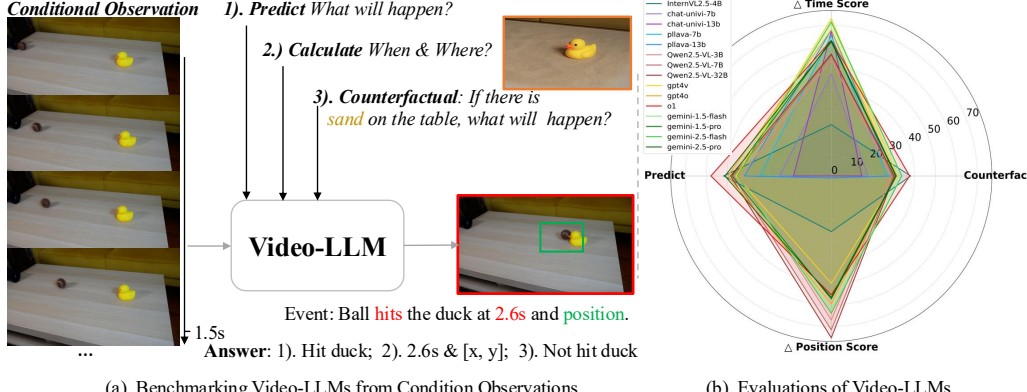

(a). Benchmarking Video-LLMs from Condition Observations          (b). Evaluations of Video-LLMs

Figure 1: (a) The three-tiered challenges of CoPhyBench. The Video-LLM must first predict what will happen from conditional observations, calculate *when* and *where* events will occur, and a hypothetical counterfactual setting. In this example, the ball will hit the duck at 2.6 seconds, under a constant velocity assumption, but may slow down and not reach the duck if there are sands on the table. (b) Performance of 15 state-of-the-art Video LLMs on CoPhyBench. While some models show moderate success in event prediction, their performance on spatiotemporal and counterfactual reasoning remains substantially lower, indicating persistent challenges in physics-based inference.

els can move beyond mere pattern recognition and instead apply physical equations and causal logic to unseen scenarios.

We curate a high-quality dataset comprising $1,300$ diverse real-world physics videos-question-answer pairs across three tasks: *Prediction*, *Calculation(Time and Position)*, and *Counterfactuals*. The questions span various physical problems, including mechanics and optics. The Video-QA pairs are organized into three tasks that vary in visual complexity and reasoning difficulty. Each QA instance is grounded in observable physical events, follows a consistent task protocol, and is cross-validated by annotators with sufficient physics knowledge to ensure correctness and clarity.

In summary, our work introduces a novel benchmark that assesses whether Video-LLMs can observe, calculate, and reason about physical phenomena from partial video input. We combine conditional prediction, quantitative estimation, and counterfactual inference in a unified framework. Our CoPhyBench comprehensively evaluates physical reasoning capabilities and highlights key challenges that current models must overcome. Our key contributions are:

- We propose a novel physics video benchmark based on a conditional event structure that evaluates a model's ability to reason about physical events from visual observations.

- We introduce a hierarchical framework that evaluates a model across three levels: prediction (*what*), physical calculation (*when, where*), and counterfactual reasoning (*what if*).

- We provide comprehensive benchmark results on state-of-the-art Video-LLMs, highlighting current limitations in physical understanding and motivating future research directions.

## 2 RELATED WORKS

### 2.1 PHYSICS-RELATED GENERATION BENCHMARKS

The video generation models, which aim towards real-world simulators, are gaining interest from the community. Recent works have proposed video generation benchmarks to evaluate how well the video generation model follows physics laws. Recent works such as VideoPHY (Bansal et al., 2024) and PhyGenBench (Meng et al., 2024) aim to assess physical common sense through video generation tasks. These benchmarks evaluate whether models can synthesize physically plausible scenes involving dynamics such as collisions, occlusion, object permanence, fluid interactions, and

force-induced trajectories. However, most of these datasets are either synthetic or prompt-driven. Physics-IQ (Motamed et al., 2025), PISA (Li et al., 2025), and Morpheus (Zhang et al., 2025a) incorporate real physical experiments to assess generative models. However, without encoding real-world physical knowledge, generation models may struggle to synthesize physically plausible videos. CoPhyBench introduces a diagnostic benchmark grounded in real-world video data, explicitly designed to disentangle causal, computational, and counterfactual reasoning, offering deeper insight into a model's physical reasoning ability.

## 2.2 Physics-related Understanding Benchmarks

Image or language-based QA benchmarks (Johnson et al., 2017; Lu et al., 2022) only focus on object relations, not dynamics or calculation. Despite the many video benchmarks (Hu et al., 2025; Li et al., 2024b; Patraucean et al., 2023; Mangalam et al., 2023; Zhou et al., 2024; Wu et al., 2024; Fang et al., 2024; Zhao et al., 2025; Ning et al., 2023) proposed to evaluate the semantic alignment with Video-LLMs, the qualitative capability of the Video-LLMs to reason from the physical world remains unclear. Recently, PhysBench (Chow et al., 2025) is the first comprehensive benchmark to evaluate vision-language models of physical world understanding. However, it remains biased by common-sense priors from the superficial associations that LLMs acquire, leaving Video-LLMs' true visual understanding and physics-based reasoning untested. ComPhy (Chen et al., 2022) and DynSuperCLEVR (Wang et al., 2024) are based on synthetic data, thus limiting their scope to evaluating Video-LLMs for real-world reasoning. Our real-data benchmark departs from PhysBench-VQA by introducing conditional observations, calculation-based question-answers, and organizing tasks into a fine-grained reasoning hierarchy.

## 2.3 Video-LLMs

By learning a cross-modal connectioner to bridge the video and powerful LLMs (Grattafiori et al., 2024; Touvron et al., 2023; Yang et al., 2024), Video-LLMs (Maaz et al., 2023; Zhang et al., 2023; Lin et al., 2024; Li et al., 2024c; Xu et al., 2024; Li et al., 2024a; Bai et al., 2025) have significantly improved the performance of video understanding. Yet, these models emphasize high-level semantic understanding (*e.g.*, human actions and social events), rather than the underlying physical dynamics of the scene (*e.g.*, reasoning about forces, collisions, momentum, or causal consequences of physical interactions.). Thus, this paper benchmarks several the state-of-the-art Video-LLMs on CoPhyBench to assess their capabilities towards video physical reasoning.

## 3 CoPhyBench

To assess whether video-based large language models (Video–LLMs) can reason about physical phenomena from partial observations, we introduce **CoPhyBench**—the **C**onditional reas**o**ning **Phy**sical Video **Bench**mark. We first present a high-level overview of the benchmark in Section 3.1, outlining its core goals and design principles. Section 3.2 provides the detailed statistics of CoPhyBench. Section 3.3 then formalizes the underlying task of conditional physics reasoning, which challenges models to predict both qualitative and quantitative properties of future events from incomplete visual evidence. Section 3.4 defines the evaluation metrics used to assess model performance. Section 3.5 outlines the data curation process.

### 3.1 Overview of CoPhyBench

Recent progress in physical question answering (PhysQA) has produced benchmarks that test whether models can describe or explain physical events in videos. However, these datasets typically expose the model to the entire video and emphasize high-level common-sense reasoning. Yet they do not evaluate whether a model can predict what will happen before it occurs, compute precise numerical outcomes, or reason correctly under counterfactual modifications.

CoPhyBench is designed to fill this gap. It introduces a new challenge setting - *Conditional-Video Physics Reasoning* - in which models are shown only the frames up to a designated switch frame. Then, they must reason about the unseen future based solely on this partial evidence. Unlike prior PhysQA datasets, CoPhyBench directly targets three complementary aspects of physical inference,

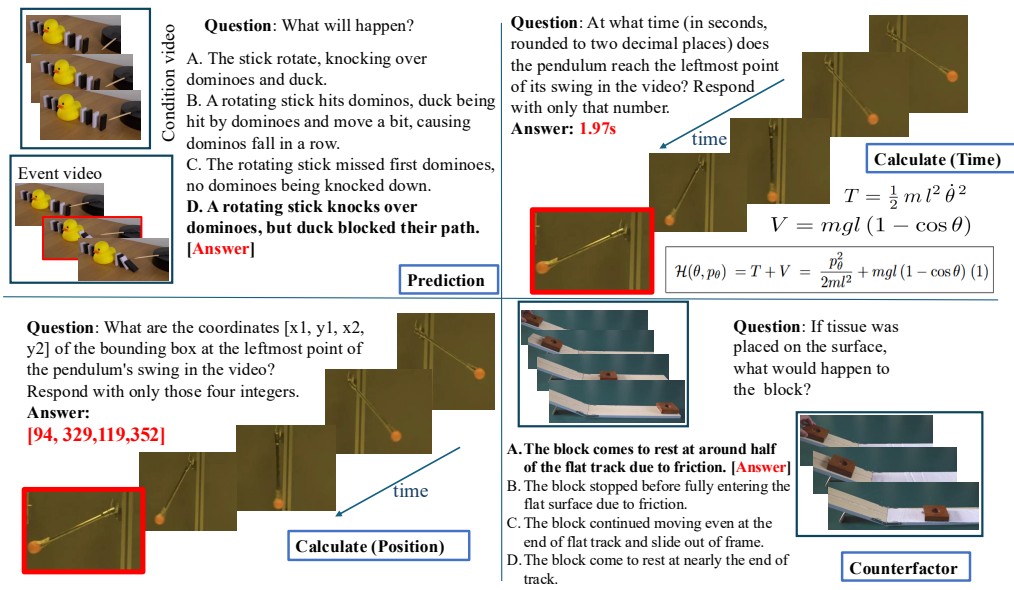

Figure 2: Examples from CoPhyBench, illustrating the three core task types: (Top-left) Causal Prediction: Given only the condition frames, the model must infer what will happen next. In this case, it must determine the duck's motion direction, which depends on subtle visual cues such as initial orientation and scene setup. (Top-right and Bottom-left) Physical Calculation: Models are asked to estimate physical quantities like the time (in seconds) when a pendulum reaches its left-most point, or the bounding box coordinates of the pendulum at that moment. These questions require extracting visual measurements and applying physical equations (*e.g.*, conservation of energy, angular motion). (Bottom-right) Counterfactual Reasoning: Models must predict what would happen under a hypothetical change: here, the addition of a tissue layer alters surface friction, and the model must reason how it would affect the block's final position. This tests whether the model encodes generalizable physical principles rather than surface-level patterns. Each task reflects a distinct dimension of physical reasoning, ranging from intuitive causality to quantitative computation and causal generalization.

as illustrated in Fig 2: (i) temporal prediction, by hiding future frames and requiring models to predict what will happen next (*how*); (ii) quantitative calculation, by requiring precise predictions of when and where the event occurs (*when and where*); (iii) counterfactual reasoning, by testing whether predictions adjust appropriately under hypothetical changes (*what if*). The statistics of CoPhyBench are presented in Figure 3.1, more detailed information is provided in the apppendix.

**Phenomenon coverage.** Our CoPhyBench spans a broad range of physical phenomena grounded in classical mechanics and beyond—including simple sliding motion in kinematics, object collisions in linear dynamics, and pendulum motion in rotational systems. This diversity ensures that the benchmark reflects various real-world physical behaviors. These scenarios are carefully curated to support the three tasks in CoPhyBench: prediction, physical calculation, and counterfactual reasoning, each requiring different levels of perceptual grounding and physics-based inference.

**Blind-QA.** As illustrated in Figure 2, without the conditional videos, the model can not derive the answer since it has no cues of the objects, such as motion direction, velocity or the physical interaction with other objects, and cannot do calculation precisely or reasoning with altering conditions.

## 3.2 DETAILED STATISTICS OF CoPhyBench

The CoPhyBench dataset comprises videos from three primary sources. As illustrated in Figure 3, approximately 40.1% (93 videos) are self-collected real-world physics experiments, ensuring controlled conditions and high-quality annotations. Additionally, 14.2% (33 videos) originate

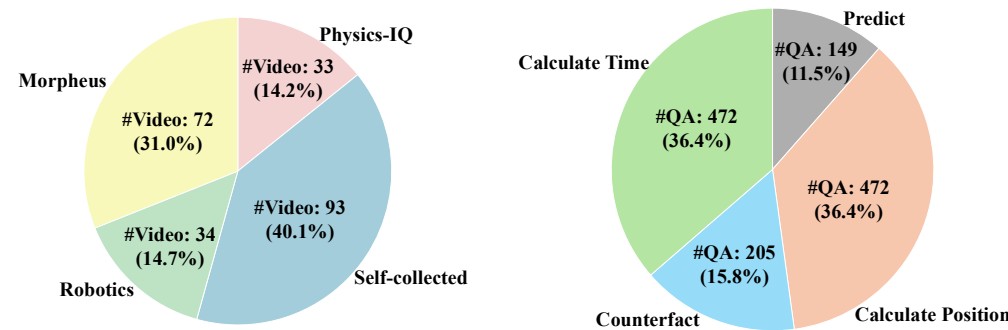

Figure 3: Data sources in CoPhyBench.          Figure 4: Reasoning types in CoPhyBench.

from the Physics-IQ dataset (Motamed et al., 2025), 31.0% (72 videos) are re-annotated clips from the Morpheus benchmark (Zhang et al., 2025a), and 14.7% (34 videos) are from Open X-Embodiment (O'Neill et al., 2024). Figure 4 presents the task distribution within the dataset. It encompasses four types of reasoning tasks: counterfactual reasoning (15.8%, 205 instances), calculating event time (36.4%, 472 instances), calculating event position (36.4%, 472 instances), and causal prediction (11.5%, 149 instances). This balanced distribution ensures comprehensive evaluation across multiple physical reasoning dimensions.

### 3.3 CONDITIONAL VIDEO PHYSICS REASONING

**Events and the Switch Frame.** An *event* refers to any visually salient state transition driven solely by the system's internal dynamics after a designated switch time $t_{\text{switch}}$, such as a collision, the apex of a projectile, or the moment an object comes to rest.

To represent this structure formally, we divide each video into two segments: the *condition clip* $C$, containing all frames up to and including the switch frame,

$$C = \{ \mathbf{I}_t \mid t \le t_{\text{switch}} \}, \tag{1}$$

and the *event clip* $E$, containing all subsequent frames,

$$E = \{ \mathbf{I}_t \mid t > t_{\text{switch}} \}. \tag{2}$$

During inference, only the condition clip $C$ is available to the model; the event clip $E$ is withheld and serves as the prediction target.

**Formalization of Dynamics.** To provide a structured account of physical behavior from partial video observations, we model each scene as a Hamiltonian system (Feynman, 1999), a classical mechanics framework that expresses motion in terms of energy. In this formulation, the total energy of a system is represented by the Hamiltonian $H(q, p, t)$, which combines kinetic energy $T(p)$ and potential energy $V(q)$. Here, $q(t) \in \mathbb{R}^d$ denotes the generalized coordinates (e.g., positions), and $p(t) \in \mathbb{R}^d$ is the corresponding momenta (e.g. for position coordinate x, the corresponding momenta is $p = m\dot{x}$), forming the system state

$$\mathcal{S}(t) = \big(q(t), p(t)\big) \in \mathbb{R}^d \times \mathbb{R}^d. \tag{3}$$

The dynamics evolve according to the Hamiltonian equations:

$\dot{q} = \frac{\partial H}{\partial p}, \dot{p} = -\frac{\partial H}{\partial q}$, with the Hamiltonian defined as: $H(q, p) = T(p) + V(q)$, where $m_i$ is the mass of the $i$-th object or coordinate dimension.

This formulation enables structured physical reasoning by decomposing the task into two interrelated steps. First, the model must perform **symbolic identification**: from visual evidence in the condition clip $C$, it infers the functional forms of the system's energy terms, $T(p)$ and $V(q)$. For instance, it may recognize a quadratic potential $V(q) = \frac{1}{2}kq^2$, or identify uniform motion by detecting the absence of external forces.

Second, the model conducts **parameter estimation**: once the symbolic structure is determined, it must estimate a set of physical parameters $\theta = m_i, k, \gamma, \ldots$, which may include, for example,

masses ($m_i$), stiffness constants ($k$), damping coefficients ($\gamma$), or other quantities depending on the system. These parameters define the specific dynamics encoded by the Hamiltonian and must be inferred from visual observations.

For example, in simple cases, only a few frames are sufficient for parameter estimation due to the low degrees of freedom in typical physical systems. Each frame provides a snapshot of the system's state, and finite differences across successive frames approximate time derivatives. Two frames yield an estimate of velocity ($\dot{q}$), while three or more enable the estimation of acceleration ($\ddot{q}$), which is essential for recovering parameters related to force, mass, or stiffness. The model may require longer condition clips to infer physical parameters in more complex scenarios reliably.

If the symbolic structure and physical parameters are both correctly identified, the model can reconstruct the Hamiltonian $H$ and simulate the system forward in time from the switch frame. It can then predict the system state $\mathcal{S}(t)$ for any $t > t_{\text{switch}}$, and thereby infer key characteristics of the target event, including its type, timing, and spatial location.

**Factual Prediction Queries (Casualty and Calculation).** Each video is paired with human-written queries that target specific physical phenomena to test this capability. For example, in a projectile scene, the model may be asked: *"What is the time and horizontal position of the projectile's highest point?"* A correct answer must (i) confirm that the apex occurs (Casualty), (ii) estimate the time of the apex $t_{\text{apex}}$ (Calculation), and (iii) provide its spatial location $\mathbf{x}_{\text{apex}}$ in the image plane (Calculation). Similar queries are constructed for collisions, rebounds, pendulum extrema, and other dynamic events.

By evaluating these responses, the benchmark assesses whether a model can infer symbolic dynamics, estimate physical parameters, and simulate forward—all *from partial visual input alone*. Viewed through this Hamiltonian lens, CoPhyBench challenges models to go beyond recognizing physical behavior and express and solve the governing equations.

**Counterfactual Formalization.** Beyond predicting what will happen given the observed initial conditions, a physically grounded agent must also reason about what *would* happen if conditions were altered. To that end, we introduce a counterfactual reasoning component that builds directly on the model's inferred Hamiltonian dynamics from the factual setting. This extension tests whether a model can internally modify its representation of the physical system and simulate how outcomes would change under hypothetical interventions.

Given the original condition clip $C$ and the symbolic and parametric Hamiltonian $H$ inferred before, we define a counterfactual condition

$$\tilde{C} = \text{ALTER}(C; \Delta\theta, \Delta q, \Delta p), \tag{4}$$

where $\Delta\theta$ denotes perturbations to system parameters (e.g., doubled mass, halved spring constant), and $(\Delta q, \Delta p)$ refers to changes in the initial state (e.g., higher release height, reversed velocity). Importantly, $\tilde{C}$ is not shown as a video; the modification is conveyed through natural language descriptions such as: *"If the incline angle were reduced by half..."* or *"Suppose the ball's mass is doubled at launch..."*

**Counterfactual Prediction Queries.** The model is then asked to predict what would happen in the modified system $\tilde{H}$, obtained by updating $H$ with $\theta \leftarrow \theta + \Delta\theta$ and applying the shifted initial state $(\Delta q, \Delta p)$. The query is a multiple-choice question, generated initially by GPT and then revised by expert annotators for physical plausibility and linguistic clarity.

## 3.4 EVALUATION METRICS

**Scoring for causal and counterfactual reasoning.** We evaluate performance on causal prediction and counterfactual reasoning using top-1 accuracy over the evaluation split:

$$\text{Acc} = \frac{1}{|\mathcal{D}|} \sum_{i \in \mathcal{D}} \mathbf{1}[\hat{a}_i = a_i^*], \tag{5}$$

where $a_i^*$ denotes the ground-truth choice for clip $i$, and $\hat{a}_i$ is the model's predicted answer.

**Scoring for calculations.** We measure the error in event time and 2D position for quantitative predictions. Let $\hat{t}, \hat{x}, \hat{y}$ denote the model's outputs, and let $t^*, x^*, y^*$ denote ground-truth values.

The normalized errors are defined as:

$$\text{err}_t = \frac{|\hat{t} - t^*|}{T_{\text{clip}}}, \qquad \text{err}_{xy} = \frac{\sqrt{(\hat{x} - x^*)^2 + (\hat{y} - y^*)^2}}{R_{\text{max}}}, \tag{6}$$

where $T_{\text{clip}}$ is the total clip length, and $R_{\text{max}}$ is the larger of the frame width or height (in pixels). If either error exceeds the validity threshold (i.e., a normalized error greater than 1) or if the answer string is malformed, the sample receives zero credit. Otherwise, we assign the per-sample score:

$$s_i = (1 - \text{err}_t); \quad s_j = (1 - \text{err}_{xy}), \tag{7}$$

where $s_i$ denotes the Relative Time Score while $s_j$ is Relative Position Score.

### 3.5 DATASET CONSTRUCTION & ANNOTATION

**Data sources.** The dataset is constructed from four main sources. We re-annotate publicly available videos from Physics-IQ (Motamed et al., 2025) and Morpheus (Zhang et al., 2025a), which are two physics-based video generation benchmarks. We also reformulate the well-controlled robotics videos from Open X-Embodiment (O'Neill et al., 2024). We select sequences that match our taxonomy and enrich them with switch-frame annotations and event-level numeric labels. Note for Physics-IQ (Motamed et al., 2025), they already provide the switch frame. We also capture 93 high-frame-rate clips (60fps, 1080p) using a modular in-house rig. These recordings cover controlled setups including ramps, pendulums, and elastic and inelastic impacts in a physics laboratory. More information and illustrations are provided in the appendix.

**Annotation protocol.** For each clip, an initial caption is generated using GPT-4o (Achiam et al., 2023) and then reviewed by two trained annotators with sufficient physics knowledge to correct any factual or conceptual inaccuracies. These verified captions are then used to instantiate prompt templates that generate the four answer candidates of *Prediction* and *Counterfactor* queries. Annotators iteratively refine the resulting question-answer pairs until a blind GPT-based QA model fails to answer them correctly. This iterative process ensures that the prompts cannot be solved by relying solely on pretraining knowledge. L2 annotations are collected using a custom in-house interface that allows annotators to scrub through the video, mark the event frame, and record the object's 2D position $(x^*, y^*)$ via bounding box placement.

## 4 EXPERIMENTS

This section evaluates the performance of current Video–LLMs on COPHYBENCH and highlights the uniqenesses of our benchmark compared to prior efforts such as PhysBench (Chow et al., 2025).

### 4.1 EXPERIMENTAL SETUP

Our evaluation was conducted under four task configurations: (a) Prediction, (b) Time Calculation, (c) Position Calculation, and (d) Counterfactor. We evaluate open-source Video-LLMs like Chat-Univi (Jin et al., 2024), PLLaVA (Patraucean et al., 2023), InternVL2.5 (Chen et al., 2024) and Qwen2.5-VL (Bai et al., 2025). We also test the performance of closed-source Video-LLMs, such as Gemini (Team et al., 2023) and GPT4 (Achiam et al., 2023). All models are fed 8 frames, which take the image resolution as their default setting. For the other settings, we follow PhysBench (Chow et al., 2025). We provide more detailed information in the supplementary materials.

### 4.2 BLINDQA COMPARISONS OF PHYSBENCH *vs* COPHYBENCH

We evaluate PhysBench (Chow et al., 2025) with a BlindQA protocol on the validation set—that is, the models receive only the textual portion of each example to answer the question directly. As Table 1 shows, both Gemini-1.5-Pro (Team et al., 2023) and GPT-4V (Achiam et al., 2023) achieve high accuracy with only textual inputs. We present some examples of QA pairs in which the answer can be derived easily from only the text parts.

- "Question": "<video>What happens to the liquid column in the flask when hot water is poured? A. The liquid column

Table 1: BlindQA and Non-BlindQA results on PhysBench-*val* and CoPhyBench.

| | | PhysBench | | | | |
|---|---|---|---|---|---|---|
| **Benchmark** | **Model** | **Property** | **Relationships** | **Scene** | **Dynamics** | **Avg.** |
| | Random | 25.00 | 25.00 | 25.00 | 25.00 | |
| PhysBench | GPT-4V (Blind) | 37.04 | 62.50 | 54.84 | 47.17 | 49.63 |
| | GPT-4V | 66.67 | 70.83 | 38.71 | 66.04 | 60.74 |
| | Gemini-1.5-Pro (Blind) | 33.33 | 70.83 | 25.81 | 64.15 | 50.37 |
| | Gemini-1.5-Pro | 70.37 | 83.33 | 41.94 | 66.04 | 64.44 |

| | | CoPhyBench | | |
|---|---|---|---|---|
| **Benchmark** | **Model** | **Predict** | **Counterfact** | **Avg.** |
| | Random | 25.00 | 25.00 | 25.00 |
| CoPhyBench | GPT-4V (Blind) | 28.38 | 25.37 | 26.87 |
| | GPT-4V | 45.64 | 31.71 | 38.67 |
| | Gemini-1.5-Pro (Blind) | 21.62 | 28.29 | 24.95 |
| | Gemini-1.5-Pro | 47.65 | 31.22 | 39.43 |

> falls. B. The liquid column rises. C. The liquid column
> remains unchanged. D. The flask breaks." "Answer": "B. The
> liquid column rises."

- "Question": "<video>What change occurs to the size of the
  ice cube in the video? A. Decreasing. B. Increasing.
  C. No change. D. Increasing first and then decreasing.",
  "Answer": "A. Decreasing."

As shown in the listed examples, even without watching the video, we find LLMs are easy to know that the liquid column rises when pouring hot water into the flask. Also, it is easy to know that the size of ice cube will decrease in normal condition. These findings indicate that the results of PhysBench (Chow et al., 2025) are highly susceptible to language bias. In contrast, our CoPhyBench shows negligible language bias, and the BlindQA results further demonstrate its reliability.

### 4.3 Visual Prediction *vs* Counterfactuals

*Prediction* measures models' capability to reason temporal evolutions while *Counterfact* requires the capability to reason based on both visual conditions and world physics knowledge. As illustrated in Table 2, all evaluated Video-LLMs perform better on visual prediction than counterfactual reasoning. Once the task flips a scene condition, all the same models collapse. For example, GPT-4V (Achiam et al., 2023) loses 30.5%, showing that its strong visual prior does not convert into counterfactual understanding. While even the strongest reasoning model GPT4-o1 (Achiam et al., 2023) still only achieves 37.56%, under-performing its performance on the conditional prediction tasks. Such comparisons indicate that current Video-LLMs are still far from using physical reasoning principles under visual conditions.

### 4.4 Time Calculations *vs* Position Calculations

We further compare model performance on two quantitative sub-tasks: predicting the time at which a physical event occurs (Relative Time Score) versus estimating its spatial position (Relative Position Score). Although both require numerical reasoning, the error patterns reveal a clear discrepancy. Most models perform slightly better on temporal estimation, suggesting that timing can often be inferred from visual cues such as frame counts or motion rhythms. In contrast, spatial prediction demands finer-grained geometric reasoning, *e.g.*, understanding object trajectories, depth, and interactions, which remains more challenging.

As illustrated in Table 2, GPT-4V (Achiam et al., 2023) and Gemini-1.5-Flash (Team et al., 2023) demonstrate the strongest performance on the Relative Time Calculation task, with GPT-4V leading at 73.20%. Their abilities to leverage video frame dynamics allow them to infer event timing more accurately than other architectures. GPT-4o and GPT-o1 also perform well, scoring 67.92% and

Table 2: **Evaluation results for Video-LLMs.** The performance of the evaluated Video-LLMs is illustrated in the table. More results are included in the supplementary materials. *Average* is the mean across the four metrics: Prediction, Time, Distance, and Counterfactual.

| Model | Prediction | Relative Time Score | Relative Position Score | Counterfact | Average |
|---|---|---|---|---|---|
| Random Choice | 25.00 | - | - | 25.00 | - |
| InternVL2.5-4B (Chen et al., 2024) | 51.68 | 23.87 | 25.88 | 37.07 | 34.63 |
| Qwen2.5-VL-3B (Bai et al., 2025) | 43.62 | 53.75 | 67.07 | 28.78 | 48.31 |
| Qwen2.5-VL-7B (Bai et al., 2025) | 46.98 | 56.11 | 71.61 | 29.76 | 51.12 |
| Qwen2.5-VL-32B (Bai et al., 2025) | 48.32 | 56.73 | **75.32** | 30.24 | 52.65 |
| Chat-UniVi-7B (Jin et al., 2024) | 24.83 | 47.43 | 0.00 | 17.56 | 22.45 |
| Chat-UniVi-13B (Jin et al., 2024) | 18.12 | 67.34 | 0.00 | 14.63 | 25.02 |
| PLLaVA-7B (Xu et al., 2024) | 34.23 | 65.35 | 0.95 | 26.83 | 31.84 |
| PLLaVA-13B (Xu et al., 2024) | 41.61 | 67.91 | 0.95 | 29.27 | 34.94 |
| GPT-4V (Achiam et al., 2023) | 45.64 | **73.20** | 49.93 | 31.71 | 50.12 |
| GPT-4o (Achiam et al., 2023) | 49.66 | 67.92 | 59.60 | 30.73 | 51.98 |
| GPT-o1 (Achiam et al., 2023) | **57.72** | 63.13 | 55.26 | **37.56** | 53.42 |
| Gemini-1.5-Flash (Team et al., 2023) | 44.97 | 71.53 | 62.41 | 33.66 | 53.14 |
| Gemini-1.5-Pro (Team et al., 2023) | 47.65 | 63.06 | 57.29 | 31.22 | 49.81 |
| Gemini-2.5-Flash (Team et al., 2023) | 44.97 | 71.74 | 63.86 | 33.82 | **53.60** |
| Gemini-2.5-Pro (Team et al., 2023) | 51.01 | 62.41 | 56.44 | 30.88 | 50.19 |

63.13% respectively. Among the open-source models, Qwen2.5-VL-7B achieve midrange results of 56.73%, lightweight architectures like InternVL2.5-4B lag noticeably behind at 23.87%.

In contrast, relative position calculation reveals a broader performance spread. Qwen2.5-VL-32B excels with a 75.32 score, indicating strong geometric reasoning and depth inference capabilities. The Gemini series also demonstrates strong spatial perception, with scores of 63.86% (Gemini-2.5-Flash) and 62.41% (Gemini-1.5-Flash). However, despite being unified image–video models that excel at position reasoning, specialized video models like Chat-UniVi-7B and Chat-UniVi-13B obtain score 0, while PLaVA-7B (0.95%) and InternVL2.5-4B (25.88%) show only minimal capability, underscoring the challenge of precise spatial reasoning.

Overall, while most multimodal models effectively capture temporal dynamics, only a subset, primarily Qwen2.5-VLs and the Gemini Flashs, demonstrate comparable prowess in spatial tasks. Bridging this gap will require future Video-LLMs to incorporate more sophisticated spatiotemporal reasoning mechanisms.

### 4.5 MODEL SIZE ANALYSIS

As illustrated in Table 2, model scale follows a consistent "scaling law" trend: as parameter counts increase, performance generally improves across all tasks. For example, Chat-UniVi's performance rises from 22.45% at 7B to 25.02% at 13B, while PLaVA's score increases from 31.84% to 34.93%. For small-scale models (Qwen2.5-VL-3B), scaling up brings even larger performance gains, for example, Qwen2.5-VL's relative time score increases by 5.80% from 3B to 7B, but it increases only marginally by 2.90% from 7B to 32B, indicating diminishing returns in reasoning beyond a certain model parameter scale. These trends suggest that scaling up model size encounters a bottleneck to improve conditional physics reasoning. Effective physics reasoning in Video-LLMs likely requires better grounding mechanisms and task-specific supervision rather than just more parameters.

## 5 CONCLUSION

We introduce COPHYBENCH, a Conditional ReasOning Physics Benchmark designed to evaluate the ability of Video-LLMs to reason about real-world physical events. By incorporating prediction, physical calculation, and counterfactual reasoning tasks grounded in real video observations, CO-PHYBENCH moves beyond semantic alignment to test deeper physics-based understanding. Our dataset contains 1, 300 carefully verified video-question-answer pairs covering diverse phenomena and reasoning difficulties. Benchmarking results reveal that while existing models perform moderately well on visual prediction, they struggle with precise computation and counterfactual reasoning—limitations that prior benchmarks fail to uncover due to language biases and lack of computational grounding. Thus, we hope COPHYBENCH will serve as a standard benchmark to motivate future large multimodal models to advance towards physics reaonsing in real world.

ETHICS STATEMENT

All experiments used publicly available models and datasets under their respective licenses. Any additional data were collected by the authors without involving human subjects or personally identifiable information. The work does not include human- or animal-subjects research, privacy-invasive analyses, discriminatory practices, or potentially harmful dual-use experiments.

REPRODUCIBILITY STATEMENT

All technical details, such as experiment settings, evaluation protocols and implement instructions are detailed described in Sec. 4.1 of the paper and Sec. B.1 of the supplementary to ensure reproducibility. All datasets used in this paper are publicly accessible. The code to reproduce our experiments will be released after the review process.

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

# APPENDIX

In the appendix, we first declare LLM usage in paper writing, then present more detailed information and visualization of representative examples the proposed COPHYBENCH in Section A, and implementation details in Section B, the construction details of COPHYBENCH in Section C, and discuss the potential limitations and broader impact in Section D.

## THE USE OF LARGE LANGUAGE MODELS

In this paper, Large Language Models (LLMs) were used exclusively for polishing the manuscript, including improving writing style, enhancing readability, and correcting grammatical errors. LLMs were not employed for research purposes such as literature retrieval, idea generation, or discovery. All methodological proposals, experimental designs, analyses, and conclusions were developed without the involvement of LLMs.

## A   BENCHMARK EXAMPLES

### A.1   OVERVIEW OF BENCHMARKS

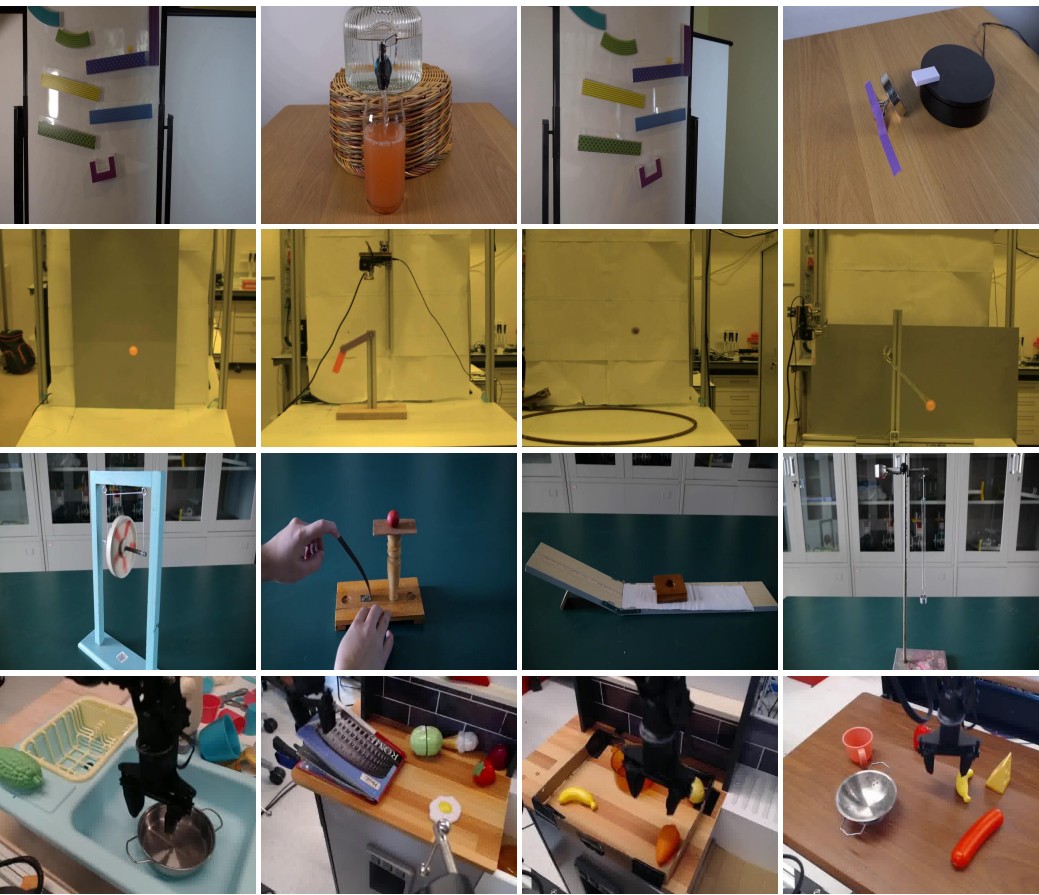

Figure 5: Overview from our COPHYBENCH benchmark. In each column, the four rows (top to bottom) show Physics-IQ (Motamed et al., 2025), Morpheus (Zhang et al., 2025a), Self-Collected Videos, and Robotics (O'Neill et al., 2024).

In addition to reformulating videos from Physics-IQ (Motamed et al., 2025), Morpheus (Zhang et al., 2025a) and Open X-Embodiment (O'Neill et al., 2024), we collected a portion of our dataset in a real-world setting under the supervision of a certified school physics teacher.

These videos were recorded in a middle school physics laboratory using standard experimental apparatus. The setups capture a variety of classical mechanics phenomena such as falling motion, inclined plane dynamics, frictional effects, and energy transfer through sound waves. Each video consists of two segments: the condition clip showing the initial setup, and the event clip capturing the outcome. By systematically varying initial conditions, such as surface materials, object mass, and applied forces. We constructed diverse and challenging physical scenarios suitable for evaluating fine-grained reasoning abilities in Video-LLMs. Figure 6 presents visualizations of the collected examples. The complete dataset will be publicly released.

## A.2 VIDEO INFORMATION FOR CoPhyBench

Table 3: Frame rate and resolution by source

| Source | FPS | Resolution |
|---|---|---|
| Physics-IQ | 30 | $912 \times 512$ |
| Morpheus | 30 | $640 \times 512$ |
| Self-Collected | 30 | $912 \times 512$ |
| Robotics | 30 | $640 \times 512$ |

Table 3 summarizes the video sources in CoPhyBench, reporting the frame rate (FPS) and spatial resolution for each dataset. For *Self-Collected* videos, we resampled the videos to 30 FPS. When evaluating Video-LLMs, we use the input videos in their default format.

## A.3 DESCRIPTION OF SELF-COLLECTED PHYSICAL LABORATORY DATA

The self-collected data contains eight sceneros: `Ball_bouncing`, `Ball_fall_to_tube`, `Mass_on_Spring`, `Ramp_Ball`, `Ramp_Block`, `Ramp_Block_with_friction`, `Rolling_Wheel`, `Single_Pendulum` and `Sound_wave_energy_transfer`.

The detailed description of the scnenroy in Figure 6 are as followed:

- **Ball_bouncing.** An elastic small ball is thrown upward. For different conditions, the initial velocity of the throw is varied, making the highest point the ball reaches the key event. Additionally, despite sharing the same acceleration (i.e., gravity), different initial velocities result in varying times for the ball to contact the desktop. This setup demonstrates projectile motion principles, where the maximum height depends on initial velocity, and the total time of flight varies accordingly, commonly used in physics experiments to study energy conservation and kinematics.

- **Ball_fall_to_tube.** A metal spring is activated, causing it to flip away a wooden board and trigger the red ball to fall. By varying the condition—specifically, the ball's position on the wooden board—different outcomes are produced, such as the ball landing inside or outside the vertical tube.

- **Mass_on_Spring.** A mass stretches the spring under the force of gravity. When the mass is released from different initial heights, it possesses different amounts of kinetic and potential energy. That is, under varying initial conditions (mass positions), the mass reaches different maximum heights at different times. This setup demonstrates how changing the starting position influences both the energy and the timing of the system's motion.

- **Ramp_Ball.** A ball is released from different positions on the inclined ramp and undergoes projectile motion. As the initial release height of conditions changes, the point where the ball lands after its projectile motion also changes.

- **Ramp_Block.** A sliding block is placed at different heights on an inclined ramp to vary the speed conditions. By changing the starting position, the sliding block achieves different speeds and kinetic energies, allowing it to undergo uniform deceleration on the flat plane. The different

releasing conditions result in different stopping points. Namely, the block may stop in the middle of the board, at the end, or even slide off completely.

- **Ramp_Block_with_friction.** Similar to `Ramp_Block`, a sliding block is placed at different heights on an inclined ramp to vary the speed conditions. The flat plane is set up with different friction coefficients, *e.g.*, towels, fabrics, or papers of varying roughness levels. When the block is released from the same height, it will have the same speed upon entering the flat plane motion. With different friction resistance conditions, the block will stop at different positions.

- **Rolling_Wheel.** Initially, the wheel is at rest at the lowest point, with a rope wound around its rim. Under different initial conditions, the rope is wound to different lengths. After the wheel is released, the differing rope lengths cause the wheel to take different times to return to the lowest point.

- **Single_Pendulum.** The pendulum is displaced to various initial angles and released to swing strictly within a vertical plane. The release is from rest, so gravitational potential energy is converted into kinetic energy during the motion. Because the launch angle sets the initial height, different initial angles produce different times for the bob to return to the lowest point.

- **Sound_wave_energy_transfer.** A small ball (ping-pong ball) is suspended by a string and initially at rest. A tuning fork is struck with varying force, then translated at constant speed to make gentle contact with the ball. The fork's vibrations transfer energy to the ball and push it away. Stronger strikes impart larger vibration amplitudes, producing greater contact impulse and causing the ball to reach a higher maximum height.

## B    DETAILED MODEL STAMPS OF GPTS AND GEMINIS

This section lists the model names and API stamps used in the experiments, adhering to official naming conventions and providing documentation links for reproducibility.

| Model Name | API Model Stamp | API Reference Link |
|---|---|---|
| GPT-4V | `gpt-4-turbo` | GPT-4-Turbo |
| GPT-4o | `gpt-4o` | GPT-4o |
| GPT-o1 | `o1` | GPT-o1 |
| Gemini 1.5 Flash | `gemini-1.5-flash` | Gemini 1.5 Flash Docs |
| Gemini 1.5 Pro | `gemini-1.5-pro` | Gemini 1.5 Pro Docs |
| Gemini 2.5 flash | `gemini-2.5-flash` | Gemini 2.5 Flash Docs |
| Gemini 2.5 Pro | `gemini-2.5-pro` | Gemini 2.5 Pro Docs |

Table 4: API model stamps and official reference links for the GPT family and the Gemini family (1.5 Flash/Pro, 2.5 Flash/Pro).

## C    CONSTRUCTION DETAILS OF COPHYBENCH

We present the construction details and processing pipeline for COPHYBENCH.

For the question construction in the *Prediction* task, we select questions from 16 general prediction templates listed in Table 5. For answer and distractor generation, we use GPT-4o with the instructions in Figure 7.

For *Calculation* task, we use GPT-4o with the instruction prompt below to generate a question prompt for evaluated Video-LLMs.

Table 5: Template for *Predict* questions in CoPhyBench

**Prediction Question Templates**

- Based on the current visual evidence, what is the most likely next event?
- Given the observable physical setup, what will happen next?
- What is the most plausible outcome that follows this scene?
- What will most likely occur in the next moment, based on what is shown?
- Predict the next physical event using visible cues.
- According to the observed interactions, what happens next?
- From the visual setup, infer the immediate consequence.
- Based on the objects' motion and position, what is about to happen?
- Given the current frame sequence, what is the expected outcome?
- What physical change is most likely to occur next?
- What is the most probable next step in this sequence of events?
- Use visual evidence to anticipate what will happen next.
- Considering the current physical forces in action, what will happen?
- What event is likely to follow based on this setup?
- From the observed behavior, what is the next expected motion or reaction?
- What is the logical continuation of this physical interaction?

**Time Calculation:** Given the label {label}, write a single-sentence question that asks a viewer to watch the video, calculate when the {label} event occurs in seconds (rounded to two decimal places), and require to respond with only that number.

**Position Calculation:** Given the label {label}, write a single-sentence question that asks a viewer to watch the video, determine the bounding box [x1, y1, x2, y2] where the {label} event occurs, and require to respond with only those four integers.

For *Counterfact* task, the instruction prompts to generate question-answers are shown in Figure **??**.

## D  LIMITATIONS AND BROADER IMPACT

**Limitations.**  While CoPhyBench provides a rigorous benchmark for evaluating conditional physics-based reasoning in Video-LLMs, it focuses on short and controlled video segments to enable calulation of "when" and "where". This may not fully reflect the diversity and complexity of real-world environments.

**Boarder Impact.** From a broader perspective, CoPhyBench promotes a shift from superficial association to deeper physical reasoning, with potential applications in robotics, simulation, scientific discovery, and education. However, the ability to predict and intervene in physical scenarios also raises safety concerns—especially if such models are deployed in decision-critical domains without reliable uncertainty estimation. We encourage future research to combine physical reasoning with grounded real-world validation and interpretability mechanisms.

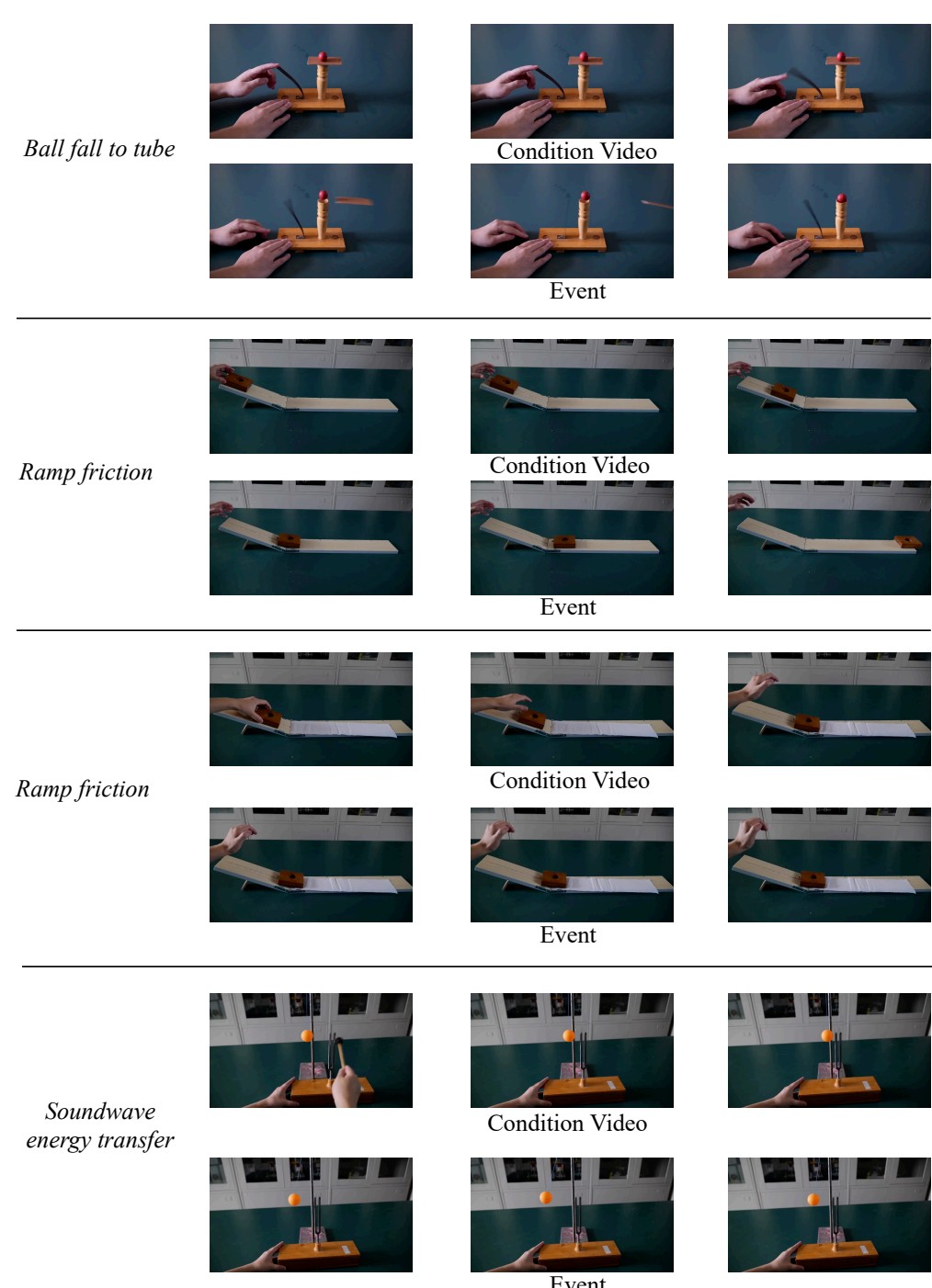

Figure 6: Examples from our CoPhyBench benchmark. Each block shows a real-world physical event segmented into a condition video (top row) and its corresponding outcome (bottom row). The leftmost labels describe the physical scenario, including "Ball fall to tube", "Ramp friction" (with two different surface conditions), and "Soundwave energy transfer". These diverse cases reflect the benchmark's coverage of physical scenarios.

Take "[EVENT_LABEL]" as reference. Watch the objects in the video frames carefully and analyze the physical "
"phenomena depicted in the scene.
"You are required to generate one high-quality and challenging multiple-choice questions. "
"The question must include four answer options (A–D), with only one being correct. All choices must be "
"textually plausible, physically consistent, and mutually confusing. "
"The correct answer must be clearly supported by either visual observation or physically grounded reasoning, "
"depending on the task level. "
"Do not include any instruction-related phrasing in the question text itself. "
"\n\nPrediction under Partial Observation:\n"
"You have access to the full video, but the model will only be shown the first part of the video (i.e., the "
"initial frames leading up to the event). "
"Based on the final state and results, design a question that requires the model to predict what will happen "
"next using observable physical cues. "
"The question must demand causal reasoning about the outcome, based on the visible setup. "
"All answer choices should be physically plausible continuations based on observable objects' motion, interaction, or state changes, "
"but only one should match the actual outcome in the entire video. "
"Avoid reflecting on information not present in the first segment. "
"\n\nAnswer Choice Construction Requirements:\n"
"- **Choice A must be exactly the caption/description of the first video frame.**\n"
"- All four choices must be textually plausible, grammatically consistent, and mutually confusing.\n"
"- The distractors (Choices B, D) must be plausible physical behaviors or state changes of objects closely related to the effector(s) present in the video frames, ”

" acting as confusers.\n"
"- Do not allow answers to be guessed via commonsense, statistical priors, or language patterns.\n"
"- Use outcome-based descriptions involving motion, direction, force response, timing, or position change—"
"not static traits like size or color.\n"
"- Avoid using object labels or simple names as cues for correctness.\n"
"\nFinal Review Instructions:\n"

"After generating the questions and answers, carefully revise them to ensure:\n"
"1) the correct answer cannot be identified without seeing the video or applying physics-based reasoning,\n"
"2) all distractors are textually convincing and physically valid but incorrect,\n"
"3) no option can be ruled out through language shortcuts alone.\n"
"\nOutput Format (strict):\n"

"Question: <insert prediction question based only on early frames>\n"
"A) <choice A – caption of the first frame>\n"
"B) <choice B –distractor>\n"
"C) <choice C – correct answer based on the actual final outcome>\n"
"D) <choice D –distractor>"

Figure 7: Prompt for GPT-4o to generate prediction questions, correct answers, and distractors.

-------- Video Input --------
You are provided with the FULL video clip, limited to eight representative frames. Carefully observe visual details, setups, and physical relationships in the scene.
-------- Question-Writing Task --------
Your job: WRITE ONE challenging counterfactual multiple-choice question (MCQ) that tests a model's ability to predict how the outcome would change if exactly ONE clearly visible visual condition or physical attribute in the video were modified.
In the question stem, briefly state the altered visual condition or attribute based on observable features (e.g., "Suppose the surface is slightly more sloped," "Suppose the block is marginally closer to the edge," or "Suppose the left object's position is a bit higher").
Provide 4 answer options (A–D) with only one correct answer (C). All options must be physically plausible, mutually confusing, and derive from the altered setup.
The correct answer must rely purely on counterfactual physical reasoning grounded in visual evidence—avoid hints that allow the answer to be deduced simply by language patterns.
-------- Writing Rules --------
The stem must precisely name ONE change (visual condition or physical attribute), avoiding specific numerical values; use qualitative expressions only (e.g., "slightly," "marginally," "a bit higher," or "closer").
Describe outcomes using motion, direction, force response, timing, or position change—do not rely on color, size, or labels as cues.
Keep language concise; do not include meta-instructions or extraneous commentary.
-------- Final Review Checklist --------
✓ The correct answer requires both video observation and physical reasoning.
✓ All distractors are reasonable under some physical explanation but incorrect for the altered scenario.
✓ No option can be eliminated through language shortcuts alone.
-------- Strict Output Format --------
Question: <counterfactual question stem describing the single altered visual condition>
A) <plausible but incorrect>
B) <plausible but incorrect>
C) <correct answer>
D) <plausible but incorrect>

Figure 8: Prompt for GPT-4o to generate counterfactual questions, correct answers, and distractors.

