# OpenReview forum: "CoPhyBench: Benchmarking Physical Reasoning from Conditional Video Observation"
_ICLR.cc/2026/Conference — ICLR 2026 Conference Withdrawn Submission_

### Official Review · Reviewer_D5Wq · 2025-10-31

**Soundness:** 3
**Presentation:** 2
**Contribution:** 2
**Rating:** 2
**Confidence:** 4

**Summary:**

The paper present CoPhyBench, a new benchmark designed to evaluate the physical reasoning capabilities of video vlms. The insight of this paper is that existing benchmarks often test superficial common-sense correlations, which can be solved by language priors alone. CoPhyBench addresses this by let the video vlms only see an initial video segment and must reason about the unseen outcome. It evaluates models on three hierarchical tasks: 1) Causal Prediction (what will happen), 2) Physical Calculation (when and where), and 3) Counterfactual Reasoning (what if). The dataset includes 1,300 question-answer pairs grounded in 232 real-world videos. Key findings show that while models are adequate at prediction, they struggle significantly with precise calculation and counterfactuals.

**Strengths:**

- The insight of this paper is reasonable, i.e. existing benchmarks often test superficial common-sense correlations, which can be solved by language priors alone.
- The paper provides compelling evidence (the BlindQA) that CoPhyBench is not susceptible to the language-based biases found in prior work. Models performed near-randomly with text-only input, proving that visual grounding is essential to solve the tasks.

**Weaknesses:**

- The paper's core motivation is hard to follow. It claims (line 40) that a prior benchmark "suffers from language biases," but the results supporting this claim aren't shown until Section 4.2. This forward reference is confusing and weakens the introduction. More importantly, the authors don't clearly explain why their new "Causal Prediction" task doesn't have the same language bias problem. They need to provide more direct evidence or explanation to justify this key distinction.

- The "Formalization of Dynamics" in Section 3.3 is out of place. The paper strongly implies that models must follow a "symbolic identification and parameter estimation" chain to succeed. Is this really the only path? It's plausible that a model could find the right answer through other, less formal reasoning shortcuts, without needing to estimate every parameter and solve the equations. This section seems to prescribe one specific, complex method for a task that might not always require it.

- The data construction and annotation process is too vague. The paper just says "two trained annotators with sufficient physics knowledge". This isn't enough to establish reliability. What were their qualifications? In the paper I can hardly know the benchmark quality.

**Questions:**

See in weekness.

---

### Official Review · Reviewer_N1wM · 2025-10-31

**Soundness:** 3
**Presentation:** 1
**Contribution:** 2
**Rating:** 2
**Confidence:** 4

**Summary:**

This paper introduces a benchmark for testing Video-LLMs’ physical reasoning ability. They collect a dataset of 1300 videos from various sources, and test three kinds of tasks, given a short conditioning video: a) the predict task involves stating in text what will happen next, b) the calculation task ask the model to estimate physical variables like the time at which an event will occur and the spatial location, and c) the counterfactual task involves reasoning about alternative futures as a result of some modification in the input stimuli.

**Strengths:**

1. Understanding whether large video models have physical reasoning capabilities and testing in which areas they can improve is an important and open problem. The paper does a decent job of combining various physical reasoning tasks into a suite of benchmarks.
2. It was encouraging that the test introduced here is able to separate various models in terms of their performance. The key findings that GPT-o1 is better than GPT-4v at prediction but not at calculation, and the Gemini series of models emerging as the strongest model was interesting.

**Weaknesses:**

1. The authors seem to use the word hypothetical and counterfactual interchangeably. As per the causal inference literature these are not exactly the same (see Pearl’2000). A hypothetical involves reasoning about plausible future states of the world, and a counterfactual involves reasoning about how the current state of the world might change if something in the past were modified. I think the task introduced by the authors qualifies as counterfactual so they can stick to that word throughout.
2. In general, I think the paper does a poor job at positioning it’s proposed benchmark relative to what already exists. The authors mention that CoPhyBench should have negligible language bias compared to Physbench, but only report two isolated examples on BlindQA. It could be an important issue, but because the observation hasn’t been sufficiently developed in this paper, it’s seeming like the authors are merely pointing out some minor flaw in Physbench. There are a few other benchmarks such as intphys2, physion and inflevel which the authors do not mention. While these are meant for intuitive physics understanding, they can potentially be repurposed for testing video-LLMs.
3. The size of the dataset, and the models tested are somewhat limited, which makes it hard to make confident conclusions. For instance, prior benchmarks like PhysBench evaluated 75 models, and has 10000 queries, but CoPhyBench only has 1300 videos across 3 tasks.
4. Implications are not discussed in enough detail, mainly because of the limited set of models tested. Given the results on the benchmark, how should the field take the next step?
5. There may be a way of combining existing benchmarks to test for the calculation and counterfactual task. For instance, one could use TAO-Amodal [2] to track points on objects and compute physical quantities like velocity. As these are based on off the shelf-tracker they can be scaled up to many videos, and we don’t really need to reannotate videos for this task. And for testing counterfactual understanding one could use CausalVQA[3]. So, I’m not sure why one would use CoPhyBench as opposed to just testing models on a combination of these existing benchmarks. Maybe the authors should clarify why their benchmark is preferable.


Typos & writing improvement:

L196: mporal prediction, should be temporal?
Figure 2: what is meant by counterfactor
Paragraph in L207 is abruptly introduced
L411: why use “counterfact” as shorthand? It makes things confusing, as if this is a new term.
L404 grammar issue: “we find LLMs are easy to know”

I think the whole section on formalization seems unnecessary. The goal of the paper is to introduce the benchmark and present empirical results. A lot of the theory introduced for physical dynamics is already well known and doesn’t add much value in reintroducing it here because it’s not a contribution of the paper.

References:
[1] Judea Pearl, Causality: Models, Reasoning, and Inference. (2000)
[2] TAO-Amodal: A Benchmark for Tracking Any Object Amodally
[3] CausalVQA: A Physically Grounded Causal Reasoning Benchmark for Video Models

**Questions:**

As I mentioned towards the end of the listed weaknesses, my main concern is that this benchmark is not really adding anything new to the suite of benchmarks that already exist. Can the authors clarify more why this benchmark is useful?

---

### Official Review · Reviewer_wZLx · 2025-11-01

**Soundness:** 2
**Presentation:** 2
**Contribution:** 3
**Rating:** 4
**Confidence:** 2

**Summary:**

COPHYBENCH is a benchmark for evaluating Video-LLMs on physics-based reasoning from real-world videos. It includes three tasks: (1) causal future prediction, predicting future events from initial observations, (2) physical calculation, estimating quantitative variables such as time and position, and (3) counterfactual reasoning, answering “what if” questions beyond what is directly seen. The dataset contains over 200 real-world physics videos and 1,300 verified QA pairs across various kinematic and dynamic phenomena. Results show that while models perform moderately well on causal prediction, they struggle with quantitative estimation and counterfactual reasoning, revealing limits in physics-grounded understanding.

**Strengths:**

1. The paper tackles an important and timely topic, assessing physics-grounded reasoning in Video-LLMs.
2. The idea of evaluating models from conditional video observations is intuitive and practical, reflecting how humans reason about physical dynamics from partial information.
3. The dataset is grounded in real-world videos, covering diverse physics phenomena (kinematics, dynamics, optics).
4. The results, if valid, are insightful in showing that current Video-LLMs succeed at causal prediction but struggle with quantitative estimation and counterfactual reasoning, exposing clear limits in their physics-based understanding.

**Weaknesses:**

1. The conclusions are not fully convincing. Since no model performs well on counterfactual tasks, it is hard to tell whether the issue lies in the task design or the model limitations. The results could reflect either underspecified questions or genuinely weak model reasoning, and the paper should include more examples or human baselines to clarify this.
2. The benchmark’s underlying principles are not well defined. Each task makes sense individually, but it is unclear whether they collectively form a coherent framework or just cover arbitrary question types. It is also uncertain whether these questions are meant to be complete or complementary to other benchmarks for evaluating physical understanding.
3. The writing, particularly in Section 3.3, is confusing and lacks coherence between paragraphs. Some claims are overstated, such as insisting that the model “must” solve symbolic identification.

**Questions:**

1. How can readers be confident that the benchmark questions are well-posed and solvable rather than underspecified? Presenting human evaluation results could help demonstrate that the tasks are interpretable and have unambiguous answers.
2. How should zero or near-zero model scores be interpreted? Could they reflect linguistic misunderstanding rather than failure of physical reasoning?
3. What principle connects the three tasks? Are they intended as representative axes of physics understanding or illustrative examples? How is the benchmark’s overall scope of “physics understanding” defined, and what are its current limitations?
4. How are symbolic identification and parameter estimation linked to the benchmark tasks? For example, do counterfactual tasks test the ability to identify symbolic relations or estimate parameters? How does the Hamiltonian framework guide the task design or interpretation?

I would raise my rating if the authors ensure task soundness through human validation, clarify the conceptual framework, and establish the scope and limitations of the benchmark clearly.

---

### Note · Authors · 2025-11-24

**Comment:**

Dear Reviewers,

Thank you for taking the time to give feedback on our work.  We will take it into account as we revise it for future submissions.

- Authors

**Withdrawal Confirmation:**

I have read and agree with the venue's withdrawal policy on behalf of myself and my co-authors.